# Relationship between Injuries and Attention-Deficit Hyperactivity Disorder: A Population-Based Study with Long-Term Follow-Up in Taiwan

**DOI:** 10.3390/ijerph19074058

**Published:** 2022-03-29

**Authors:** Yo-Ting Jin, Miao-Ju Chwo, Chin-Mi Chen, Shi-Hao Huang, Yao-Ching Huang, Chi-Hsiang Chung, Chien-An Sun, I-Long Lin, Wu-Chien Chien, Gwo-Jang Wu

**Affiliations:** 1Department of Nursing, College of Medicine, Fu-Jen Catholic University, New Taipei City 24205, Taiwan; jinyoting@gmail.com (Y.-T.J.); 071471@mail.fju.edu.tw (M.-J.C.); 128135@mail.fju.edu.tw (C.-M.C.); 2Department of Nursing, National Taipei University of Nursing & Health Sciences, Taipei 11230, Taiwan; 3Graduate Institute of Medical Sciences, National Defense Medical Center, Taipei 11490, Taiwan; gwojang@yahoo.com; 4Department of Chemical Engineering and Biotechnology, National Taipei University of Technology (Taipei Tech), Taipei 10608, Taiwan; hklu2361@gmail.com (S.-H.H.); ph870059@gmail.com (Y.-C.H.); 5Department of Medical Research, Tri-Service General Hospital, National Defense Medical Center, Taipei 11490, Taiwan; g694810042@gmail.com; 6School of Public Health, National Defense Medical Center, Taipei 11490, Taiwan; 7Taiwanese Injury Prevention and Safety Promotion Association, Taipei 11490, Taiwan; 8Big Data Research Center, College of Medicine, Fu-Jen Catholic University, New Taipei City 24205, Taiwan; 040866@mail.fju.edu.tw; 9Department of Public Health, College of Medicine, Fu-Jen Catholic University, New Taipei City 24205, Taiwan; 10Department of Computer Science and Engineering, Tatung University, Taipei 104327, Taiwan; cyberpaul@gm.ttu.edu.tw; 11Graduate Institute of Life Sciences, National Defense Medical Center, Taipei 11490, Taiwan; 12Department of Obstetrics and Gynecology, Tri-Service General Hospital, Taipei 11490, Taiwan

**Keywords:** injuries, risk, attention-deficit hyperactivity disorder, disabilities, children

## Abstract

Objective: To investigate the association between various injuries and attention-deficit hyperactivity disorder (ADHD) and distinguish ADHD from non-ADHD with regards to risk of various injuries among children in Taiwan. Method: Using the data from the National Health Insurance Research Database, we selected a total of 1802 subjects under the age of 18 who were diagnosed with ADHD as well as an additional 7208 subjects as a comparison group. Results: Compared with children who were not diagnosed with ADHD, children diagnosed with ADHD were more likely to intentionally injure themselves. During the school year, ADHD children were injured less frequently than were non-ADHD children on traffic-related incidents. The adjusted hazard ratio of injury for the ADHD children was 2.493 times higher than that of comparison subjects. The ADHD children had a greater length of stay and medical cost when compared to those of the non-ADHD children. Age showed a significant inverse relationship with injury. Among the ADHD children, the injury rate was evidently higher for the low-income group than for the non-low-income group. Conclusions: Age, cause of injuries, low-income household status, and school season all have a significant connection to the risk of injury for ADHD children.

## 1. Introduction

Attention-deficit hyperactivity disorder (ADHD) is a cerebral dysfunction that results in problems with concentration and impulse control in about half of the adults who were diagnosed with ADHD as child [1]. The estimated worldwide-pooled prevalence rate of ADHD among persons 18 years or younger is 5.3% [2,3]. In 2003, only 4.3% of children between the ages of 4 and 17 who were previously diagnosed with ADHD were taking medication for the disorder [4]. Injury is a major cause of death among children and adolescents throughout the world, responsible for over 875,000 deaths in children and young people under the age of 18 years each year [5]. The majority of researches indicate that children diagnosed with ADHD (“ADHD children”) have a higher rate of accidental injuries than children without ADHD (“non-ADHD children”) [4,6,7,8,9,10,11], and that without treatment, ADHD often leads to increased risk of injuries, automotive crashes, traffic citations, bone fractures, and head injuries in adolescence [12]. Researchers do not fully understand the reasons behind the increased risk of injuries among ADHD patients, but it may be that someone with ADHD is more likely to be inattentive, distracted, impulsive, or lacking in the ability to recognize the consequences of certain behaviors, compared to someone without ADHD [4].

In 2011, the prevalence of ADHD diagnoses in U.S. children (ages 4–17 years) was approximately 8%, and children affected by ADHD were more vulnerable to physical injuries such as physical trauma, accidental poisoning, burns, etc. [13]. Other studies reported ADHD prevalence rates of 7.6–9.5%, 10–20%, and 29.7% in Korea, India, and the United Arab Emirates, respectively [14,15,16]. Al Zaben FN et al. (2018) found an overall ADHD incidence of 5%, with a higher incidence in boys than in girls. They reported that the most prevalent subtype of ADHD was combined ADHD (2.7%), followed by hyperactive ADHD and inattentive ADHD (1.2% and 1.1%, respectively) [17].

The lack of long-term follow-up and comparisons among different causes of injuries in ADHD children could lead to inconsistent results. Therefore, the purpose of this study was to examine the association between ADHD and various injuries and to distinguish ADHD from non-ADHD children with regards to risk of various injuries with long-term follow-up from 2000 to 2010 in Taiwan’s medical environment. Approximately 20 years ago, the Taiwanese government initiated a single-insurer National Health Insurance (NHI) program and over 23 million people were enrolled in the National Health Insurance Research Database (NHIRD) after the program was implemented. The NHIRD allowed researchers access to a large amount of long-term data for analysis. 

## 2. Method

### 2.1. Data Source

Taiwan implemented National Health Insurance on 1 March 1995, and the health insurance coverage rate currently exceeds 99%. The NHIRD collects nationwide medical data including inpatient, outpatient, and emergency room services, and the law requires that all hospitals and clinics report all medical expenses to the Bureau of National Health Insurance on a monthly basis. Consequently, National Health Insurance information can serve as representative empirical data in medical- and health-related research fields [18]. Researchers are required to pass a detailed review by a professional peer review committee before they can use the Taiwan’s NHIRD. Because patients’ identities are encrypted in the database, this study did not infringe on patients’ privacy rights. 

In this study, we selected patients under the age of 18 who were diagnosed with ADHD based on the International Classification of Diseases Ninth Revision (ICD-9) codes 314 between January 2000 and December 2010 from the NHIRD. Figure 1 represents a flow chart illustrating the criteria of sampling the ADHD children and following up with their injuries. As of 31 December 2010 (the date upon which our data are concluded), there were 140 and 635 injuries out of 1802 ADHD children and 7208 non-ADHD children, respectively (TSGHIRB number 2-105-05-082).

### 2.2. Variable Definitions

The study design is of a cohort study. The case group consisted of minors under the age of 18 who suffered from ADHD with injury and who joined the National Health Insurance for medical treatment. The control group consisted of children who did not suffer from ADHD with injury. People in the case and control groups were matched in terms of index date, gender, and age at the ratio of 1:4. The following variables were included in the comparison: gender, age, income level (two groups consisting of patients from low-income and non-low-income families), catastrophic illness (two groups consisting of patients with/without catastrophic illness, such as cancers, an Injury Severity Score of ≥16, and rare diseases), Charlson Comorbidity Index (CCI), circumstance characteristics, injury type, injury severity, and cause of injury.

Characteristics of circumstance included season of admission, urbanization level (three levels: high, middle, and low), level of care (three levels: medical center, regional hospital, and local hospital), and department of visit (four departments: internal medicine, surgery, pediatrics, and others). 

CCI selects the first five diagnostic codes (ICD-9-CM N-Code), weighs these 5 codes according to 19 disease scoring criteria defined by Charlson, and calculates the total score. The higher the score, the greater number of complications or the more severe the diagnosis [19].

The severity of injury was measured by length of stay (days), medical cost (USD), and prognosis (survival/mortality). 

Injury types were classified into 13 groups in accordance with the ICD-9-CM (Appendix A) and the cause of injury included the intentionality of injury (Appendix B).

### 2.3. Statistical Analysis

The SPSS 20.0 statistical software was utilized in this research. The chi-square test/Fisher exact test, the independent samples t-test, and the percentage test were used to compare the differences between the ADHD children and non-ADHD children in demographics, cause of injury, CCI, and inpatient season. Furthermore, we used Cox regression to examine the injury rate of each independent variable (*p* < 0.05).

## 3. Results

Among the 1802 patients with ADHD in the baseline study, 1416 were male and 386 were female (Table 1). The average ages of ADHD children and non-ADHD children were 5.67 and 4.28, respectively. The two groups were significantly different in terms of low-income household and catastrophic illness. The number of ADHD children from low-income families was 7.77% compared with 0.74% in the non-ADHD group. The percentage of ADHD children with catastrophic illness was 15.37% compared with 3.83% in the non-ADHD group. With respect to the CCI score, there was not a significant difference between ADHD children and non-ADHD children (0.10 vs. 0.08). 

With respect to the follow-up appointments, the average age of ADHD children and non-ADHD children increased to 7.63 and 6.27, respectively. Of the ADHD children, 8.82% were from low-income families and 16.48% had catastrophic illness. The rates of low-income household and catastrophic illness in the non-ADHD group were significantly less than those of the ADHD children (Table 2).

Table 3 shows the characteristics of circumstance of children’s admission at the end of the follow-up. Both ADHD children and non-ADHD children experienced their highest hospital admission rates in the summer (28.97% and 27.39%, respectively). The ADHD children were hospitalized more frequently in high-urbanization areas (46.39%) and less frequently in low-urbanization areas (11.71%) than were their counterparts. A significantly higher proportion of the ADHD children received medication in medical centers than that of the non-ADHD children (53.77% vs. 37.40%). Among those with ADHD, 48.78% received pediatric medication, compared to 76.19% of the non-ADHD children. 

The rates of injury for ADHD children and non-ADHD children were 7.77% and 8.81%, respectively (Table 4). However, ADHD children were also injured more severely than the non-ADHD children. The average length of stay was significantly longer for the ADHD children than for the non-ADHD children (9.79 vs. 4.42 days). Significantly higher medical costs were incurred by the ADHD children compared to the non-ADHD children (USD 879.74 vs. 663.43). 

The two groups were also significantly different on the injury type (Table 5). Open wound, superficial injury, and poisoning were more frequent in ADHD children than in non-ADHD children, whereas fracture was more frequently observed in the non-ADHD children. The incidence rate of injury for the ADHD children was significantly higher than that of the non-ADHD children (814.85 vs. 440.46), as shown in Table 6. The probability of an accident happening and resulting in an injury for ADHD children was 2.493 times higher than that for non-ADHD children (*p* < 0.001).

Table 7 shows the factors of injured hospitalized ADHD children at the follow-up appointments. As these children get older, the aforementioned probability decreases by 7.5% on an annual basis (*p* < 0.001). ADHD children in low-income families also show a significantly higher probability of getting hurt than their non-low-income family counterparts (*p* < 0.001). Figure 2 further shows the cumulative risk of injury among children under the age of 18 stratified by ADHD with the log-rank test. The probability of injury for ADHD children is significantly higher than that of non-ADHD children (log-rank *p* = 0.031). 

In addition, not all patients had data available on their cause of injury because the NHIRD does not require medical institutions to provide that information. In this study, a total of 476 patients provided information on the cause of the injury; 68 were ADHD children and 408 were non-ADHD children (Table 8). Based on this information, more ADHD children were injured with other injuries (29.41%) and more non-ADHD children were injured with traffic injuries (30.64%). Generally speaking, ADHD children have a significantly higher percentage of intentional injuries (10.29%) than that of non-ADHD children (1.47%). Table 9 further shows the distribution of inpatient injury stratified by ADHD status, season of admission, and cause of injury. During the spring and fall seasons, ADHD children have a lower percentage of injury resulting from traffic incidents than non-ADHD children (7.17% vs. 30.61%, and 13.04% and 30.09%, respectively). 

## 4. Discussion

Previous studies have seldom reported the injury characteristics and severity of ADHD with long-term follow-up. This study attempted to fill in this gap by harnessing representative information from a large health insurance database. In general, ADHD children experience less severe consequences to risky behaviors and are less likely to develop prevention strategies and safety rules [20]. It is a common belief that because of difficulty with sustained attention and inability to respond appropriately to situations, ADHD children are at higher risk of injury than their healthy counterparts [21]. As expected, this study found that the overall estimate of relative risk injury among the ADHD children is 2.493 times higher than for non-ADHD children (log-rank *p* < 0.031); ADHD is positively associated with an increased risk of injuries. Moreover, the literature also found that the severity of trauma was directly associated with ADHD [22], i.e., ADHD children had more severe injuries [23]. In our study, the ADHD children had a greater length of hospital stay and higher medical costs when compared to non-ADHD-children. These findings are consistent with previous studies. Moreover, one research presents a strong link between income and rates of injury in many countries [24]. Income may result in exposure to negative physical or social environments as well as a lack of resources to purchase items necessary for safety, such as bicycle helmets [25]. In our study, ADHD children in low-income households were more frequently injured than those in non-low-income homes. 

However, there are also differences in other studies. Davidson [26] reviewed previous studies investigating the relationship between hyperactivity and injury and concluded that all of the retrospective studies reporting hyperactivity as an important risk factor for injuries were flawed and that prospective studies failed to find such association. Christoffel et al. [27] also reported that hyperactivity, impulsivity, and other behavioral disorders in children were not high-risk factors for pedestrian injury. Byrne et al. [28] indicated that preschoolers with ADHD did not significantly sustain more injuries that warranted medical treatment in an emergency department. Moreover, only one follow-up study was conducted on injured paediatric patients with ADHD in Asia; Kang et al. found that ADHD children appear to be at higher risk of injury than non-ADHD children [8]. However, the severity and disease sub-typing of injuries could not be determined in their study [8].

Analyzing the characteristics of injury yields a different result in this study. Several studies reported that the forms of injuries among ADHD children were different from those of non-ADHD children [4,29], such as fracture and intracranial and internal injury. In this study, ADHD children had a significantly higher risk of major and severe injuries involving open wound, poisoning, and superficial injury. With regard to the cause of injury, non-ADHD children actually experienced a higher percentage of unintentional injuries when compared to ADHD children. Additionally, ADHD children have a lower percentage of injuries related to traffic incidents and falling. ADHD children are unable to develop safety strategies when confronted with perceived danger [23]; however, generally speaking, ADHD children’s parents may pay extra attention to keep their children’s behavior in line, and other research points out that ADHD children’s mothers tend to be more controlling, which may explain the low percentages of injuries. 

In terms of the seasons, there is a significant difference in the risk of injury between ADHD children and non-ADHD children (*p* = 0.017 and *p* < 0.001, respectively) during the spring and fall (when schools are in session). Carla et al. [19] showed that ADHD children were injured more frequently on the street and less frequently at school than non-ADHD children were. In our study, we analyzed each season and found that ADHD children have a much lower percentage of suffering traffic-related injuries in spring and fall than they do in summer and winter. In contrast, non-ADHD children experience relatively high percentages of traffic-related injuries in all four seasons. 

Moreover, the frequency of accidents could be predicted by ADHD, gender, and age [30]. Figure 2 shows that the risk of injury is significantly higher during the first 7 years of life for ADHD children as compared to non-ADHD children. However, the number of injuries reduces drastically for ADHD children starting in year 8, whereas the number of injuries remains high for non-ADHD children. Therefore, the cumulative number of injuries are relatively the same for both groups (*p* = 0.085). There are two reasons for the reduction in the number of injuries in ADHD children with increasing age. First, ADHD children were often treated with medication to prevent injury. However, the evidence regarding prevention of injuries with medication is inconclusive [31]. Moreover, approximately 40–50% of the ADHD children did not receive any medications from 2000 to 2011 in Taiwan [32]. Second, previous studies show that different age groups of ADHD children may exhibit different symptoms and behaviors, and some may show significant improvements as they get older [3]. Consequently, the younger the ADHD children, the higher the risk of injury [8,33].

## 5. Conclusions

In this study, we discovered that while ADHD does increase the risk of injury in children, as the patients get older, the difference in the risk of injury between ADHD and non-ADHD children gradually becomes smaller or even disappears completely. The injury incidence rate, average length of hospital stays, and average medical cost for ADHD children are all higher than for non-ADHD children. Additionally, among ADHD children, the low-income group has a higher incidence of injury than the non-low-income group. In comparing various causes of injury, ADHD children may suffer a higher risk in intentional injuries than non-ADHD children, but ADHD children actually experience a lower risk of injuries related to traffic incidents and falls. These findings will help develop new protocols and diagnostic criteria so that physicians are able to identify and report ADHD associations with injuries in a timelier manner. This issue is of great importance; nevertheless, it is simultaneously important to work with the latest and most accurate information. 

Future studies should investigate any changes in the observational period (every ten years) from 2011 to 2020, in areas such as awareness, organization, and support of ADHD children in any way, etc.

## Figures and Tables

**Figure 1 ijerph-19-04058-f001:**
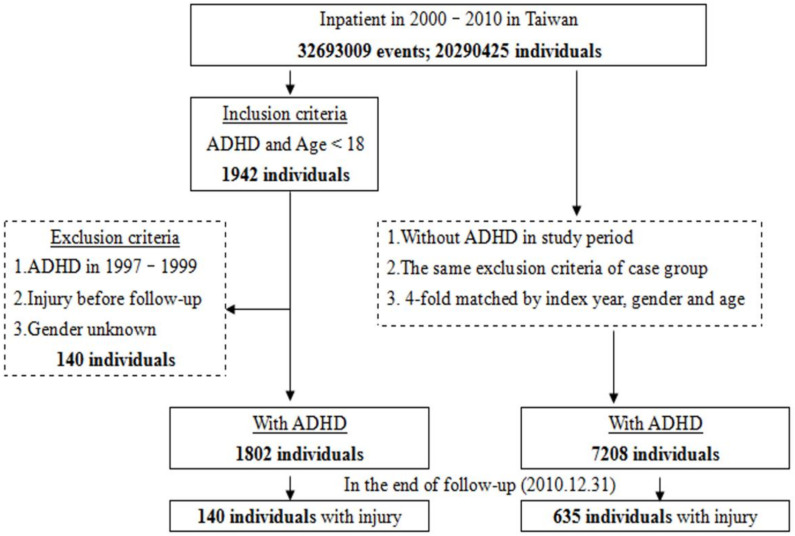
The flowchart of study sample selection from National Health Insurance Research Database in Taiwan. Note: ADHD = attention-deficit hyperactivity disorder: ICD-9-CM 314 Injury: ICD-9-CM 800-999.

**Figure 2 ijerph-19-04058-f002:**
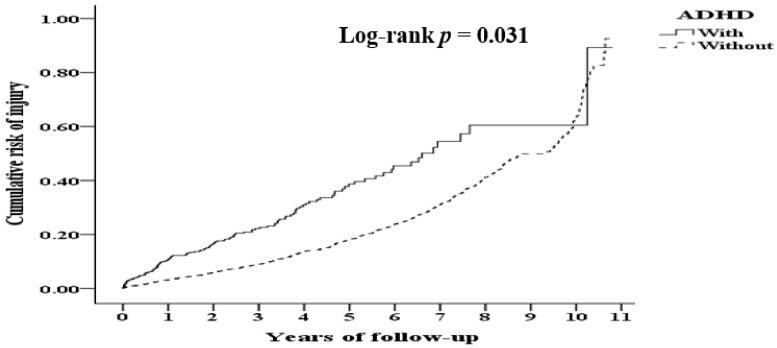
Kaplan–Meier for cumulative risk of injury among minors under the age of 18 stratified by ADHD with log-rank test.

**Table 1 ijerph-19-04058-t001:** Characteristics of hospitalized children with and without ADHD in the baseline.

Variables	With ADHD		Without ADHD		*p*-Value
	*n*	%	*n*	%	
Total	1802	100	7208	100	
Gender					0.999
Male	1416	78.58	5664	78.58	
Female	386	21.42	1544	21.42	
Age (years)	5.67 ± 4.25		4.28 ± 4.67		0.463
Low-income household					<0.001
Without	1662	92.23	7155	99.26	
With	140	7.77	53	0.74	
Catastrophic illness					<0.001
Without	1525	84.63	6932	96.17	
With	277	15.37	276	3.83	
CCI	0.10 ± 0.67		0.08 ± 0.40		0.043

*p*-value (category variable: chi-square/Fisher exact test); CCI = Charlson Comorbidity Index.

**Table 2 ijerph-19-04058-t002:** Characteristics of hospitalized child with and without ADHD at the end of follow-up.

Variables	With ADHD		Without ADHD		*p*-Value
	*n*	%	*n*	%	
Total	1802	100	7208	100	
Age (years)	7.63 ± 4.72		6.27 ± 5.42		0.129
Low-income household					<0.001
Without	1643	91.18	7135	98.99	
With	159	8.82	73	1.01	
Catastrophic illness					<0.001
Without	1505	83.52	6894	95.64	
With	297	16.48	314	4.36	
CCI	0.11 ± 0.64		0.10 ± 0.62		0.500

*p*-value (category variable: chi-square/Fisher exact test); CCI = Charlson Comorbidity Index.

**Table 3 ijerph-19-04058-t003:** Hospitalized child by characteristics of circumstance and group at the end of follow-up.

Variables	With ADHD		Without ADHD		*p*-Value
	*n*	%	*n*	%	
Total	1802	100	7208	100	
Season of admission					<0.001
Spring (March–May)	380	21.09	1822	25.28	
Summer (June–August)	522	28.97	1974	27.39	
Autumn (September–November)	483	26.80	1591	22.07	
Winter (December–Februrary)	417	23.14	1821	25.26	
Urbanization level					<0.001
High	836	46.39	2696	37.40	
Middle	755	41.90	2934	40.70	
Low	211	11.71	1578	21.89	
Level of care					<0.001
Medical center	969	53.77	2740	38.01	
Regional hospital	668	37.07	3207	44.49	
Local hospital	165	9.16	1261	17.49	
Department of visit					<0.001
Others	803	44.56	878	12.18	
Internal medicine	33	1.83	238	3.30	
Surgery	87	4.83	600	8.32	
Pediatrics	879	48.78	5492	76.19	

*p*-value (category variable: chi-square/Fisher exact test).

**Table 4 ijerph-19-04058-t004:** Hospitalized child by injury severity and group at the end of follow-up.

Variables	With ADHD		Without ADHD		*p*-Value
	*n*	%	*n*	%	
Total	1802	100	7208	100	
Injury					0.085
Without	1662	92.23	6573	91.19	
With	140	7.77	635	8.81	
Length of stay (days)	9.79 ± 13.66		4.42 ± 5.24		<0.001
Medical cost (USD)	879.74 ± 1382.42		663.43 ± 1684.71		<0.001
Prognosis					0.127
Survive	1790	99.33	7137	99.01	
Mortality	12	0.67	71	0.99	

*p*-value (category variable: chi-square/Fisher exact test).

**Table 5 ijerph-19-04058-t005:** Injured hospitalized child by injury type and group at the end of follow-up.

Variables	With ADHD		Without ADHD		*p*-Value
	*n*	%	*n*	%	
Total	140	100	635	100	
Type of Injuries					<0.001
Fracture	26	18.57	225	35.43	
Dislocation	0	0.00	6	0.94	
Sprains & strains	8	5.71	16	2.52	
Intracranial and internal injury	29	20.71	136	21.42	
Open wound	15	10.71	46	7.24	
Injury to blood vessels	0	0.00	0	0.00	
Superficial injury	16	11.43	23	3.62	
Crushing injury	1	0.71	5	0.79	
Foreign body entering through orifice	2	1.43	7	1.10	
Burns	2	1.43	30	4.72	
Injury to nerves and spinal cord	1	0.71	6	0.94	
Poisoning	12	8.57	26	4.09	
Others	28	20.00	109	17.17	

*p*-value (category variable: chi-square/Fisher exact test).

**Table 6 ijerph-19-04058-t006:** Incidence rate and adjust HR for injury among the hospitalized child at the end of follow-up.

	ADHD Children (*n* = 1802)		Non-ADHD Children (*n* = 7208)	
	*n*	%	*n*	%
Injury	140	7.77	635	8.81
Person-years	1718.11		14,416.71	
Incidence rate (per 10^5^ PYs)	814.85		440.46	
Adjust HR	2.493 *		1	
95% CI	2.019, 3.077		Reference	

Incidence rate = injury/person-years × 100,000; HR = hazard ratio, CI = confidence interval; adjusted HR: adjusted variables listed in the table; * Cox regression: *p* < 0.001.

**Table 7 ijerph-19-04058-t007:** Factors of injured hospitalized ADHD children at the end of follow-up using Cox regression.

Variables	Adjusted HR	95% CI	95% CI	*p*-Value
Gender				
Male	0.898	0.604	1.337	0.597
Female	Reference			
Age (years)	0.925	0.892	0.960	<0.001
Low-income household				
Without	Reference			
With	2.134	1.346	3.385	0.001
Catastrophic illness				
Without	Reference			
With	1.047	0.669	1.637	0.841
CCI	1.105	0.928	1.316	0.264
Inpatient season				
Spring (March–May)	Reference			
Summer (June–August)	0.959	0.565	1.626	0.876
Autumn (September–November)	1.228	0.753	2.004	0.411
Winter (December–Februrary)	0.993	0.580	1.701	0.981
Urbanization level				
High	0.823	0.523	1.296	0.401
Middle	0.729	0.463	1.148	0.172
Low	Reference			

CCI = Charlson Comorbidity Index. *p*-value (category variable: chi-square/Fisher exact test).

**Table 8 ijerph-19-04058-t008:** Injured hospitalized children by cause of injury and group at the end of follow-up.

Variables	With ADHD		Without ADHD		*p*-Value
	*n*	%	*n*	%	
Total	68	100	408	100	
Cause of injury					<0.001
Traffic injuries	13	19.12	125	30.64	
Poisoning	5	7.35	17	4.17	
Falls	14	20.59	123	30.15	
Burns and fires	0	0.00	0	0.00	
Drowning	1	1.47	1	0.25	
Suffocation	2	2.94	6	1.47	
Crushing/cutting/piercing	6	8.82	39	9.56	
Other injuries	20	29.41	91	22.30	
Suicide	4	5.88	1	0.25	
Homicide/child abuse	3	4.41	5	1.23	
Intentionality of injury					<0.001
Unintentional	61	89.71	402	98.53	
Intentional	7	10.29	6	1.47	

*p*-value (category variable: chi-square/Fisher exact test). Suicide and homicide rates were between 10 and 18 years of age.

**Table 9 ijerph-19-04058-t009:** Distribution of injury-hospitalized children stratified by ADHD status, season, and cause of injury.

Season/Cause of Injury	With ADHD		Without ADHD		*p*-Value
	*n*	%	*n*	%	
Total	68	100	408	100	
Spring (March–May)					0.017
Traffic injuries	1	7.14	30	30.61	
Poisoning	1	7.14	7	7.14	
Falls	4	28.57	33	33.67	
Burns and fires	0	0.00	0	0.00	
Drowning	0	0.00	0	0.00	
Suffocation	0	0.00	1	1.02	
Crushing/cutting/piercing	2	14.29	9	9.18	
Other injuries	4	28.57	16	16.33	
Suicide	1	7.14	0	0.00	
Homicide/child abuse	1	7.14	2	2.04	
Summer (June–August)					0.145
Traffic injuries	5	26.32	36	32.43	
Poisoning	1	5.26	4	3.60	
Falls	4	21.05	33	29.73	
Burns and fires	0	0.00	0	0.00	
Drowning	1	5.26	1	0.90	
Suffocation	0	0.00	3	2.70	
Crushing/cutting/piercing	1	5.26	8	7.21	
Other injuries	6	31.58	25	22.52	
Suicide	0	0.00	1	0.90	
Homicide/child abuse	1	5.26	0	0.00	
Autumn (September–November)					<0.001
Traffic injuries	3	13.04	34	30.09	
Poisoning	2	8.70	4	3.54	
Falls	3	13.04	35	30.97	
Burns and fires	0	0.00	0	0.00	
Drowning	0	0.00	0	0.00	
Suffocation	2	8.70	2	1.77	
Crushing/cutting/piercing	2	8.70	9	7.96	
Other injuries	7	30.43	27	23.89	
Suicide	3	13.04	0	0.00	
Homicide/child abuse	1	4.35	2	1.77	
Winter (December–February)					0.370
Traffic injuries	4	33.33	25	28.41	
Poisoning	1	8.33	2	2.27	
Falls	3	25.00	22	25.00	
Burns and fires	0	0.00	0	0.00	
Drowning	0	0.00	0	0.00	
Suffocation	0	0.00	0	0.00	
Crushing/cutting/piercing	1	8.33	13	14.77	
Other injuries	3	25.00	23	26.14	
Suicide	0	0.00	1	1.14	
Homicide/child abuse	0	0.00	2	2.27	

*p*-value (category variable: chi-square/Fisher exact test).

## Data Availability

Data are available from the NHIRD published by the Taiwan NHI administration. Because of legal restrictions imposed by the government of Taiwan concerning the “Personal Information Protection Act”, data cannot be made publicly available. Requests for data can be sent as a formal proposal to the NHIRD (http://www.mohw.gov.tw/cht/DOS/DM1.aspx?f_list_no=812 (accessed on 13 January 2022)).

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
