# Peer review of "Relationship between Injuries and Attention-Deficit Hyperactivity Disorder: A Population-Based Study with Long-Term Follow-Up in Taiwan"

_ijerph, 2022, doi:10.3390/ijerph19074058_

Round 1

Reviewer 1 Report

The topic of the article is relevant. I consider the introduction,
methods, results and discussion to be adequate. My only doubt is
regarding the Ethics Committee. Was the research approved by the
research ethics committee?

Reviewer 2 Report

The title is clear and describes the content of the manuscript.
The summary also complies with the corresponding sections.
In the introduction and justification, the background of the problem is clearly stated, as well as those issues that are not clear or require resolution.
Regarding the objectives, it is recommended to expand the main objective, as well as to formulate secondary objectives so that they describe in a more detailed way the variables that are going to be analyzed.
The results, discussion and conclusions are well structured, synthesizing extensive information that is well organized.
Include the limitations of the work, the future lines and the practices.
Update bibliographic references. The most current of the work is from the year 2016.

Reviewer 3 Report

The paper reports an interesting analysis on health data oriented to the study of the differences between children with and without Attention-Deficit Hyperactivity Disorder starting from a very important database.
First of all, it is not a probabilistic sample as all the data relating to two subgroups of interest are taken into consideration. This eliminates any risk of sampling error if we refer to the universe of individuals registered in the database which should be described more accurately. The coverage rate of this database seems very high because it currently exceeds 99%, but it would be interesting to understand this same rate referring to the observation period which could be lower. However, this data does not diminish the importance of the research carried out which lies not only in the reliability of the data, but also in the availability of context variables together with health variables.
Finally, I note that when referring to the types of injury (table 8) it is perhaps not a good idea to consider very low frequencies that could give results that are too dependent on variability and therefore less reliable to draw generalizable conclusions. Alternatively, it is possible to group into less detailed types of accidents.
Furthermore, I suggest to clarify the methodology that leads to the creation of the two observation subgroups for a better use by less expert readers of analysis techniques and research design.
Finally, the conclusions seem too synthetic and do not adequately collect the explanatory hypotheses that have been introduced in the pargraph of the results. I recommend adding some considerations on the public and social repercussions.

Reviewer 4 Report

First of all, I would like to congratulate the authors, especially for the originality of the subject matter and for the large sample of study.

However, there are some minor corrections that authors should consider to improve their work.

The introduction should be more informative, including paragraphs that should be placed in the discussion of the paper (59-61 line) and (64-66 line).

Method
There are data that appear in text and in table, my recommendation would be to put them only in one place of the work.

Tables. The format of the tables is very strange. The sample number is usually represented by a lower case n (example n=1802). I would recommend modifying some of the tables that are confusing, such as 1 and 3.

Figure 2 is faulty.

I hope that the authors will be able to make the suggested changes.

Thank you
